# Caffeic Acid Alleviates Memory and Hippocampal Neurogenesis Deficits in Aging Rats Induced by D-Galactose

**DOI:** 10.3390/nu14102169

**Published:** 2022-05-23

**Authors:** Rasa Saenno, Oabnithi Dornlakorn, Tanaporn Anosri, Soraya Kaewngam, Apiwat Sirichoat, Anusara Aranarochana, Wanassanun Pannangrong, Peter Wigmore, Jariya Umka Welbat

**Affiliations:** 1Neurogenesis Research Group, Department of Anatomy, Faculty of Medicine, Khon Kaen University, Khon Kaen 40002, Thailand; sa.aurasa@kkumail.com (R.S.); oabnithiatlakorn@kkumail.com (O.D.); tanapornanosri@kkumail.com (T.A.); soraya_km@kkumail.com (S.K.); apiwsi@kku.ac.th (A.S.); anusar@kku.ac.th (A.A.); wankun@kku.ac.th (W.P.); 2School of Life Sciences, Medical School, Queen’s Medical Centre, The University of Nottingham, Nottingham NG7 2RD, UK; peter.wigmore@nottingham.ac.uk

**Keywords:** D-galactose, neurogenesis, caffeic acids

## Abstract

Hippocampal neurogenesis occurs throughout life, but it declines with age. D-galactose (D-gal) enhances cellular senescence through oxidative stress leading to neurodegeneration and memory impairment. Caffeic acid (CA) acts as an antioxidant via decreasing brain oxidative stress. This study aims to investigate the advantages of CA in alleviating the loss of memory and neurogenesis production in the hippocampus in aged rats activated by D-gal. Fifty-four male Sprague-Dawley rats were unpredictably arranged into six groups. In the D-gal group, rats were administered D-gal (50 mg/kg) by intraperitoneal (i.p.) injection. For the CA groups, rats received 20 or 40 mg/kg CA by oral gavage. In the co-treated groups, rats received D-gal (50 mg/kg) and CA (20 or 40 mg/kg) for eight weeks. The results of novel object location (NOL) and novel object recognition (NOR) tests showed memory deficits. Moreover, a decline of neurogenesis in the hippocampus was detected in rats that received D-gal by detecting rat endothelial cell antigen-1 (RECA-1)/Ki-67, 5-bromo-2′-deoxyuridine (BrdU)/neuronal nuclear protein (NeuN), doublecortin (DCX) by means of staining to evaluate blood vessel associated proliferating cells, neuronal cell survival and premature neurons, respectively. By contrast, CA attenuated these effects. Our results postulate that CA attenuated the impairment of memory in D-gal-stimulated aging by up-regulating levels of hippocampal neurogenesis.

## 1. Introduction

Aging is reliant on a time-based system of life that is related to functional impairment, leading to a progressive decline in the organs for maintenance of their functions [1,2]. Free radicals are an important factor, which stimulate cell death in aging, especially in the brain. Neurogenesis is the generation of newborn neurons from neural progenitor cells (NPCs)/neural stem cells (NSCs) that occurs throughout life [3]. Neurogenesis in adults originates in two brain areas, including the subgranular zone (SGZ) in the hippocampus and the subventricular zone (SVZ) [4]. Oxidative stress is an important factor that induces brain aging. By contrast, an antioxidant has beneficial properties to alleviate brain aging [5,6].

D-galactose (D-gal) is an aldohexose that is enormously discovered in honey, butter, yoghurt, milk, cherry, and kiwi [7]. In normal condition, galactokinase and uridyl transferase metabolize the D-gal into glucose. After that, glucose is transferred to the glycolysis pathway or stored in adipose tissue, liver, and muscle as glycogen [8]. Excessive amounts of D-gal can generate reactive oxygen species (ROS), which results in dysfunction of mitochondria, oxidative stress, apoptosis, and inflammation, especially in neuronal cells [7]. Long-period treatment with D-gal decreases expressions of synaptic plasticity-related indicators, such as sanaptophysin and phosphorylated Ca^2^^+^/calmodulin-dependent protein kinase II [9]. Moreover, numerous investigations have confirmed that receiving D-gal induces changes of senescence-related markers in the hippocampus that leads to an increase of oxidative stress, that results in neuroinflammation, neurodegeneration and memory impairment [10,11].

Nowadays, drug therapy is widely used for management of cognitive impairment. Therefore, there is a great need for specific drugs that can be used to prevent the process involved in aging and impairments of memory and cognitive functions. Phenolic acids have earned exclusive consideration as they have steady antioxidant activity. Caffeic acid (CA), or 3,4-dihydroxycinnamic acid, is a non-flavonoid phenolic phytochemical detected in many plants, fruits, tea, and wine [12]. CA has many pharmacological functions, having antioxidant [13], anti-inflammatory [14], and anti-mutagenic effects [15], and can also reduce brain oxidative damage [16]. Koga and co-workers have studied the potential effects of CA in the nervous system in mice. The results showed that CA significantly reduces oxidative stress and activates microglia in the hippocampus. These findings suggest that CA has beneficial effects on brain health [17]. Furthermore, CA enhances the expression of brain-derived neurotrophic factor (BDNF), which encourages the synthesis of synaptic proteins, resulting in synaptic plasticity protection in hippocampal neurons. This pathway is strongly associated with learning and memory formation [18,19]. Accordingly, the main objective of the current investigation is to examine the advantages of CA in alleviating impairment of memory and neurogenesis production in the hippocampus in D-gal-activated brain aging in rats.

## 2. Materials and Methods

### 2.1. Animals

Fifty-four Sprague Dawley male rats, weighing 280-300 g, 8 weeks of age were acquired from Nomura Siam International Co., Ltd. Pathumwan, Bangkok. The rats were housed in an environment with a 12 h light-dark cycle. Three rats were allocated a plastic cage (width × length × high = 37.5 cm × 48 cm × 21 cm) at the standard room temperature in the range 23-25 °C. The rats (9 animals/group) were randomly arranged into control, D-gal, CA20, CA40, D-gal + CA20, and D-gal + CA40 groups. The study design was accepted by the Khon Kean University Ethics Committee in Animal Research (IACUC-KKU-33/63).

### 2.2. Drug Administrations

Saline solution (0.9%), 1 mL/kg and propylene glycol 1 mL/kg were administered to the control group by intraperitoneal (i.p.) injection and by oral gavage, respectively. For the D-gal group, rats were administered D-gal (50 mg/kg, Sigma Aldrich, Inc., St. Louis, MO, USA) melted in 0.9% saline solution by i.p. injection. Rats in the CA20 and CA40 groups were orally treated with CA (Sigma Al-drich, Inc., St. Louis, USA) at 20 and 40 mg/kg melted in propylene glycol, respectively. In the D-gal + CA20 and D-gal + CA40 groups, rats were administered with D-gal at the same dose as the D-gal group and CA at the same dose as the CA20 and CA40 groups. Each rat received drug administration for 8 weeks. On the first day of the treatment, each rat was administered with BrdU (Sigma Aldrich, Inc., St. Louis, MI, USA) melted in 0.9% saline solution at 250 mg/kg, with 1 mL/kg by i.p. injection.

### 2.3. Behavioral Tests

The NOL and NOR tests were performed after seventy-two hours of drug treatment. For the NOL test, rats were habituated to an arena (50 × 50 × 50 cm) for 30 min without stimuli. 24 h later, two similar objects were set in different corners of the arena. Then, rats were permitted to explore the objects for 3 min (familiarization trial). After exploring, rats were backed to their enclosures for 15 min. The arena and two objects were cleaned with 20% alcohol to wipe out odor cues. After that, one of the objects was placed at the familiar location (FL), and another one was changed to a new or novel location (NL) in the arena. Rats explored the objects in the arena for 3 min (choice trial). One day later, in the NOR test, each rat was familiarized with the arena for 30 min similar to the NOL test. Then, 24 h later, two comparable objects were laid down in different areas in the arena. Rats explored the objects for 3 min (familiarization trial). Next, rats were transferred to their enclosures for 15 min. Next, one familiar object (FO) and a new object (NO) were deposited into the arena at the familiar location. Rats explored the objects for 3 min (choice trial).

A digital camera linked to a computer program (EthoVision^®^, XT version 12, Noldus, Wageningen, Netherlands) was used to track and record the time of exploration and movement in both tests. Distance moved was noted in the habituation trial. The exploration time was counted when rats surveyed the objects by chewing, licking, sniffing, or pointing the nose towards the object from fewer than 2 cm distance [20]. Preference index (PI) was determined as a percentage of the exploration time of the NL or NO, in comparison to the total exploration time, and considered to be 50% possibility.

### 2.4. Sample Preparation

The day after NOR test, rats were sacrificed. Their brains were cut to obtain two hemispheres and immersed in a cryoprotectant containing 30% sucrose solution at 4 °C for 3 h. Then, optimal cutting temperature (OCT) compound (Themo Fisher Scientific, Darmstadt, Germany) was used to fix each hemisphere. After that, each hemisphere was promptly immersed in liquid nitrogen-cooled isopentane (Sigma-Aldrich, Inc., St. Louis, USA). Finally, the hemispheres were kept at −80 °C for immunohistochemical study.

### 2.5. Immunohistochemistry

To fix the frozen brain, 4% paraformaldehyde was used and then a freezing microtome (A.S. Science Co., Ltd., Walldorf, Germany) was used to serially section (20 µm), in coronal plane, through the length of the DG between Bregma point −2.3 to −6.3 mm. Cryoprotective buffer was used to store all sections at 4 °C. For cell proliferation investigation, every 15 sections from the complete DG were chosen using a procedure of standardized random selection to get 9 sections per brain [21,22]. The sections were reacted with anti-mouse Ki-67 antibody (1:150, NOVOCASTRA, Newcastle, UK) for 1 h followed by rabbit anti-mouse IgG Alexa Fluor 488 antibody (1:250, Invitrogen, Waltham, MA, USA, A11059) for 60 min. After that, the sections were reacted with anti-mouse rat endothelial cell antigen-1 (RECA-1) antibody (1:100, Santa Cruz Biotechnology, Dallas, TX, USA) for 60 min and then goat anti-rat Alexa Fluor 568 antibody (1:250, Invitrogen, Waltham, MA, USA, A11077) for 60 min. Finally, the sections were dyed with DAPI (1:6000, Molecular probes, Eugene, OR, USA), an additional dye, for 30 s to induce contrast of the tissue.

The immunohistochemistry of doublecortin (DCX) and bromodeoxyuridine (BrdU)/the neuronal nuclear protein (NeuN) were carried out to evaluate the number of immature neurons and cell survival correlated to mature neurons, respectively. Sections were cut at 40 µm thickness and every 8th section was selected from the entire DG to represent each brain. Finally, 9 sections per brain were used to investigate the DCX and BrdU/NeuN positive cells [21,23,24]. For the BrdU/NeuN double staining, the sections were reacted with anti-BrdU antibody (1:200, Abcam, Cambridge, UK) at 4 °C overnight, followed by incubating with Alexa Fluor 568 goat anti-rabbit antibody (1:200, Invitrogen, Carlsbad, CA, USA) for 60 min. After that, the sections were reacted with anti-NeuN (1:500, Abcam, Cambridge, UK) at 4 °C overnight, followed by rabbit anti-mouse Alexa Flour 488 antibody (1:500, Invitrogen, Carlsbad, CA, USA) for 2 h. After that, the sections were counterstained with propidium iodide (1:6000, Sigma Aldrich, St. Louis, MO, USA) for 30 s. For the DCX staining, goat polyclonal anti-DCX antibody (1:100, Santa Cruz, USA) was used to incubate the sections at 4 °C overnight, followed by antibody Alexa Fluor 488 rabbit anti-goat IgG (1:500, Molecular probes, Eugene, OR, USA) for 60 min. After incubation, propidium iodide (1:6000, Sigma Aldrich, St. Louis, MO, USA) was used to counter-stain the sections for 30 s.

### 2.6. Microscopic Quantification

A fluorescence microscope (40× objective lens, Nikon ECLIPSE 80i, Melville, NY, USA) was used to examine cells within a range of three cells at the internal edge of the DG to evaluate the numbers of positive cells of Ki-67/RECA-1, BrdU/NeuN, and DCX. The blood vessel-associated Ki-67 positive cells were considered when a positive cell was detected within a two-cell diameter interval of a RECA-1 positive blood vessel [25]. In another way, Ki-67 positive cells were considered as a non-blood vessel associated Ki-67 positive cell. Positive cell number of Ki-67/RECA-1 were multiplied by 15, and those of BrdU/NeuN and DCX positive cells were multiplied by 8 [21,23,24].

### 2.7. Statistical Analysis

GraphPad prism (Version 5.0; GraphPad Software Inc., San Diego, CA, USA) software and SPSS (Version 19.0; SPSS Inc., Chicago, IL, USA), along with demonstrated mean ± SEM, were used to analyze all statistical data. Statistical significance was revealed as *p* < 0.05. One-sample *t*-test was utilized to analyze the PI. Total exploring time, distance moved, while number of Ki-67/RECA-1, BrdU/NeuN, and DCX positive cells were evaluated using One-way ANOVA.

## 3. Results

### 3.1. D-Gal Induced Memory Impairment in NOL Test Is Reversed by CA

In the examination of habituation, the result of distance moved of all groups revealed no significant diversities (*p* > 0.05, Table 1). This result represented that locomotor activity of all rats was normal after treatment with D-gal and CA. Besides distanced moved, the result showed no significant differences in the exploration time in the familiarization trial (*p* > 0.05, Figure 1A). In the choice task, rats in the control, CA20, CA40, D-gal + CA20 and D-gal + CA40 groups surveyed the object in the novel location more than that in the familiar location (* *p* < 0.05, Figure 1B), except the D-gal group (*p* > 0.05, Figure 1B). The PIs of the control, D-gal, CA20, CA40, D-gal + CA20 and D-gal + CA40 groups showed significantly higher than 50% anticipation (Figure 2), except the D-gal group. All the data demonstrated that D-gal induced spatial memory impairment, but CA ameliorated this deterioration in a dose dependent manner.

### 3.2. D-Gal Induced Memory Impairment in NOR Test Is Reversed by CA

In the habituation trial, result of distance moved revealed no significant diversities in all groups (*p* > 0.05, Table 2), suggesting that both D-gal and CA do not impair spontaneous movement. In the familiarization trial, the exploration time of location A did not differ from location B in all groups (*p* > 0.05, Figure 3A). In the choice task, the rats in all groups, except the D-gal-treated rats, were significantly interested in the object in the familiar location less than that in the novel location (control, CA20, CA40, D-gal + CA20 and D-gal + CA40 groups: * *p* < 0.05, D-gal group: *p* > 0.05, Figure 3B). In addition, the PIs of the control, CA20, CA40, D-gal + CA20 and D-gal + CA40 groups were significantly superior to 50% anticipation (* *p* < 0.05, ** *p* < 0.01, *** *p* < 0.001, Figure 4), except the D-gal group (*p* > 0.05, Figure 4). All results demonstrated that D-gal induced loss of recognition memory, but CA attenuated this deterioration in a dose dependent manner.

### 3.3. D-Gal Reduced Cell Proliferation in the Hippocampal Neurogenesis Is Reversed by CA

Ki-67 is a nuclear protein that is expressed during cell proliferation and present in all cell cycle phases, except the G0 phase. RECA-1 is a vascular endothelial cell marker that is related to blood vessels. This study used RECA-1/Ki-67 immunofluorescence double staining to investigate blood vessel associated and non-blood vessel associated cell proliferation in the hippocampus. We found that the number of blood vessel associated Ki-67, non-blood vessel associated Ki-67 and total Ki-67 positive cells in the D-gal group were significantly less than in the control group (^#^
*p* < 0.05, ^##^
*p* < 0.01, Figure 5G–I), signifying that D-gal decreased cell proliferation in the hippocampus in any condition. Nevertheless, the rats that received only CA (20 and 40 mg/kg) demonstrated a significantly greater number of cell proliferations than those that received D-gal alone (* *p* < 0.05, ** *p* < 0.01, *** *p* < 0.001, Figure 5G–I), but this was not significantly distinct from the control group (*p* > 0.05, Figure 5G–I). These results support the view that CA, both 20 and 40 mg/kg, attenuated D-gal reduced cell proliferation in the hippocampus.

### 3.4. D-Gal Reduced Cell Survival Related to Mature Neurons in Hippocampus Is Reversed by CA

BrdU is a thymidine analog, which is used to identify DNA generation in the S phase of the cell cycle. NeuN is found during early embryogenesis in post-mitotic neuroblasts and persists in differentiating and differentiated neurons during the cell lifespan. This study used BrdU/NeuN double staining to identify cell survival related to mature neurons. The results demonstrated that BrdU/NeuN positive cells in the D-gal group were significantly lesser than in the control group (^##^
*p* < 0.01, Figure 6G), indicating that D-gal administration depleted cell survival related to mature neurons. However, BrdU/NeuN positive cells in both the CA20 and CA40 groups significantly exceeded the D-gal group (*** *p* < 0.0001, Figure 6G). Moreover, CA40 mg/kg administration up-regulated BrdU/NeuN positive cells significantly more than in the control group (^###^
*p* < 0.001, Figure 6G). Then, co-treatment with CA, both 20 and 40 mg/kg, attenuated these impairments (*** *p* < 0.0001, Figure 6G). These results revealed that CA20 and CA40 mg/kg ameliorated a reduction of cell survival related to mature neurons in the hippocampus caused by D-gal.

### 3.5. D-Gal Reduced Immature Neurons in the Hippocampus Is Reversed by CA

DCX is a microtubule associated protein, which is detected in the developing nervous system of mammalians and used as a marker of neuronal differentiation. This study used DCX immunofluorescence staining to investigate the immature neurons. The consequences of the staining showed that the D-gal group demonstrated significantly more DCX positive cells than the control group (^##^
*p* < 0.01, Figure 7G), indicating that D-gal administration depleted immature neurons. By contrast, the animals in both the CA20 and CA40 groups showed a significantly higher number of DCX positive cells than the D-gal group (* *p* < 0.05, *** *p* < 0.0001, Figure 7G). Moreover, administration with CA40 mg/kg significantly up-regulated the number of DCX positive cells so that they were higher than the control group (^####^
*p* < 0.0001, Figure 7G), however this was also found in the CA20 group. Co-administration with CA40 mg/kg attenuated D-gal induced DCX positive cell decline (* *p* < 0.05, Figure 7G). Our finding confirmed that CA40 mg/kg ameliorated D-gal reduced premature neurons in the SGZ.

## 4. Discussion

This current study confirms that CA administration reversed memory deficits in rats treated with D-gal. Rats receiving D-gal exhibited cognitive deficits and reduction in hippocampal neurogenesis, but co-administration with CA reversed these deteriorations. Moreover, our results revealed no significant differences in distance moved in all groups, indicating that locomotor activities were not affected by D-gal and CA.

Spatial memory is a memory related to spatial locations, routes or configurations. Spatial memory depends on the functions of the hippocampus [26]. In humans, spatial memory is required for remembering routes or places around a familiar city [27]. In rodents, however, it is specified to learning and remembering locations of food, water, or objects [28]. In this investigation, spatial memory was measured by the NOL test. Our results showed that prolonged D-gal administration (50 mg/kg) for eight weeks led to decrease of PIs, suggesting that D-gal can induce spatial memory impairments. Similarly, previous research has found that D-gal caused spatial memory impairments in aging rats assessed by the NOL test [29]. D-gal administration induces learning and memory impairment by elevating oxidative stress, apoptosis of neurons, synaptic protein dysfunction, microglia and astrocyte activation, and increased binding of advanced glycation end products (AGEs) to the receptor for advanced glycation end products (RAGE). D-gal also induces the releasing of inflammatory mediators, such as nuclear factor kappa-light-chain-enhancers of activated B cells (NF-κB), and causes neuroinflammation via stimulation of the transcription factor NF-κB through Ras and redox-sensitive signaling pathways, leading to cognitive deficits [30,31,32]. In the present study, CA administration, both 20 and 40 mg/kg, significantly eased D-gal-triggered decreases of PIs in a dose dependent manner.

Recognition memory is a memory related to recollection of experiences from the past, which relies on integration of the hippocampus and cortical regions [33,34,35]. The test conducted depended on the unforced propensity of rats to investigate a familiar object less than a novel one. Investigating a novel object demonstrates the utility of recognition memory [36]. This current study suggested prolonging D-gal administration led to recognition memory impairments, which were diminished by co-treatment with CA. Previous studies suggested that CA administration ameliorated learning and memory function via increases in synaptic protein expression and antioxidant functions to protect against oxidative stress induced neuronal cell damage [19,37].

It is widely accepted that adult neurogenesis is influential in learning and memory performance. The hippocampus and adjacent cortical areas located in the medial temporal lobe are essential for learning and memory processes [38]. Some proliferating cells of hippocampal neurogenesis reside in groups that are intrinsically and functionally related with a blood vessel forming a neural stem cell niche. RECA-1 is an antigen located on the endothelial cell surface, which is a component of the vascular niche. The vascular niche supports massive neurotrophic supply through production of neurotropic factors; for instance, brain-derived neurotrophic factor (BDNF), produced from endothelial cells [39,40]. One of the negative regulators of hippocampal neurogenesis is aging, which is related to significant decrease in cell proliferation, survival, and neuronal differentiation [29,41]. Age-correlated deterioration in hippocampal neurogenesis is similar to decrease in NSC population and/or activity. In aged rats, the proliferation rate is reduced, demonstrating a reduction in the activity of NSCs that are the main supporter to decrease of hippocampal neurogenesis associated with age [42]. This investigation exploited RECA-1/Ki-67, BrdU/NeuN and DCX to investigate several processes of neurogenesis, including vascular associated cell proliferation, and cell survival related to mature neurons and immature neurons, respectively [9,25,43].

It was confirmed that hippocampal neurogenesis was reduced after receiving D-gal, as demonstrated by reduction in RECA-1/Ki-67, BrdU/NeuN, and DCX positive cells in this current study. Similarly, a previous study also confirmed that Ki-67, BrdU/NeuN, and DCX positive cells are decreased after D-gal administration [9,29]. Previous studies have revealed that prolonged D-gal administration effectively impaired adult hippocampal neurogenesis by down-regulating the hippocampal cAMP-response element binding protein (CREB) and BDNF signaling pathway [44]. BDNF signaling enhances potentiated synapses strengthening and encouraging cell differentiation of NSCs into neurons [2]. The decreasing of the CREB-BDNF signaling pathway is an imperative intracellular system of age-correlated hippocampal neurogenesis deterioration [44]. According to a previous study, D-gal decreased cell proliferation by impairing antioxidant enzyme reactions and up-regulated pro-inflammatory cytokines [11]. Moreover, D-gal also decreased the rate of cell survival related to mature neurons by increasing expression levels of inflammatory mediators; for instance, interleukin-1 beta (IL-1β) and tumor necrosis factor alpha (TNFα) in the hippocampus [9]. This investigation found that co-administration with CA at 40 mg/kg was powerful in diminishing hippocampal neurogenesis deterioration produced by D-gal as demonstrated by amount of blood vessel associated cell proliferation, cell survival related to mature neurons, and immature neurons. Previous examinations have displayed that CA administration significantly enhanced antioxidant activities, resulting in down-regulation of hydrogen peroxide generation and oxidation of GSH into glutathione disulfide (GSSG), and prevented oxidative damage in nerve cells in Alzheimer’s Disease rats [18,45]. Moreover, a previous study also suggested that CA administration increased BDNF expression, which is related to synaptic plasticity, neuronal survival, and differentiation in the hippocampus and cortex of hyperinsulinemic rats [18]. Nevertheless, Liao and colleagues have reported that BDNF increased synaptic protein synthesis [46]. Synaptic proteins, including synaptophysin, PSD-95 and drebrin, perform an essential role in the pathogenesis of central nervous system diseases [47]. CA also improved synaptophysin levels in the cerebral cortex and enhanced drebrin levels in the hippocampus and cerebral cortex of hyperinsulinemic rats. Therefore, CA certainly preserved synaptic plasticity and maintained functions of neural signaling through improvement in BDNF [18]. Accordingly, CA can improve reduction in blood vessel associated cell proliferation, cell survival related to mature neurons, and premature neurons found in D-gal-produced aging. 

## 5. Conclusions

This investigation postulates that receiving D-gal leads to memory loss and hippocampal neurogenesis depletions in aging rats. Nevertheless, CA improves these deteriorations in D-gal-induced aging rats via restoring hippocampal neurogenesis. Further assessment to evaluate the hippocampal synaptic protein and antioxidant enzyme levels may benefit explanation of the mechanism of CA on memory and hippocampal neurogenesis in aging rats activated by D-gal.

## Figures and Tables

**Figure 1 nutrients-14-02169-f001:**
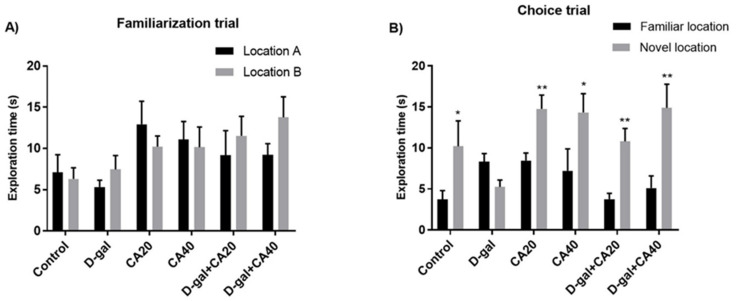
The NOL test after drug administration showed the exploration time as mean ± SEM. (**A**) The rats in all groups did not show any significant differences exploring between two objects (*p* > 0.05). (**B**) In the choice task, significant differences of the exploration time between the novel and familiar objects were detected in the control, CA20, CA40, D-gal + CA20 and D-gal + CA40 groups (* *p* < 0.05, ** *p* < 0.01), but were not observed in the D-gal group (*p* > 0.05).

**Figure 2 nutrients-14-02169-f002:**
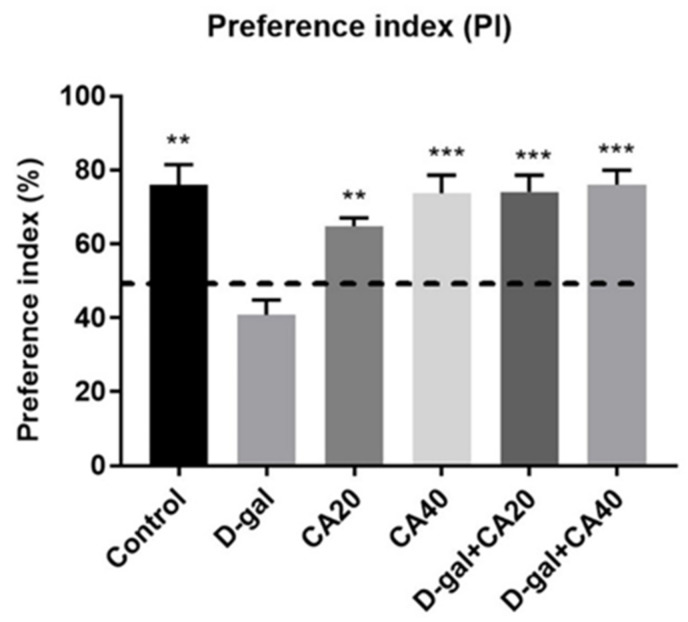
The NOL test after drug administration showed the preference index (PI) as mean ± SEM. The PIs of the control, CA20, CA40, D-gal + CA20 and D-gal + CA40 groups significantly varied from 50% possibility (** *p* < 0.01, *** *p* < 0.001), but did not show in D-gal group (*p* > 0.05).

**Figure 3 nutrients-14-02169-f003:**
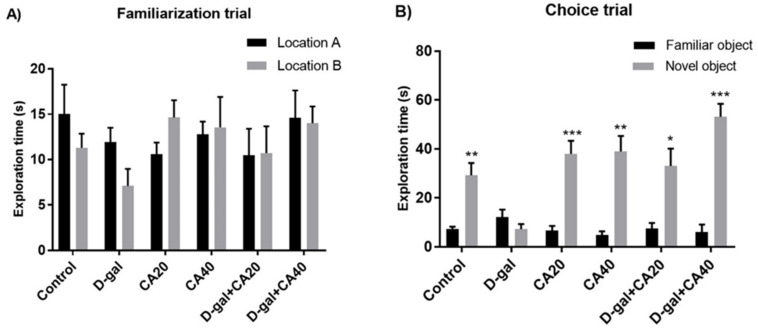
The NOL test after drug administration showed the exploration time as mean ± SEM. (**A**) No significant variations were demonstrated in exploring objects between location A and B in all groups detected in the familiarization trial (*p* > 0.05). (**B**) In the choice trial, the control, CA20, CA40 and D-gal + CA20 and D-gal + CA40 groups were interested in the object in the novel location more than that in the familiar location (* *p* < 0.05, ** *p* < 0.01, *** *p* < 0.001), but this was not observed in the D-gal group (*p* > 0.05).

**Figure 4 nutrients-14-02169-f004:**
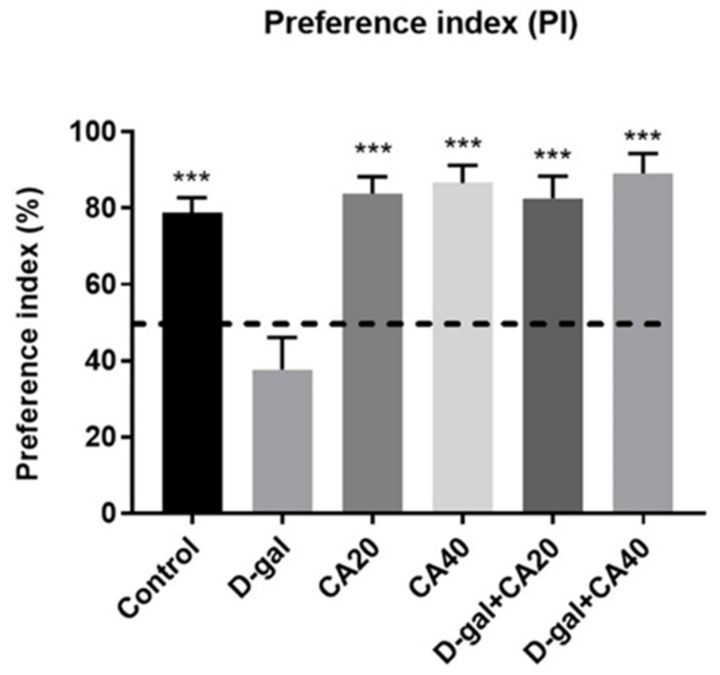
After drug administration, the data of the NOL test showed the preference index (PI) as mean ± SEM. The PIs of the control, CA20, CA40, D-gal + CA20 and D-gal + CA40 groups were significantly superior to 50% anticipation (*** *p* < 0.001), except for the D-gal group (*p* > 0.05).

**Figure 5 nutrients-14-02169-f005:**
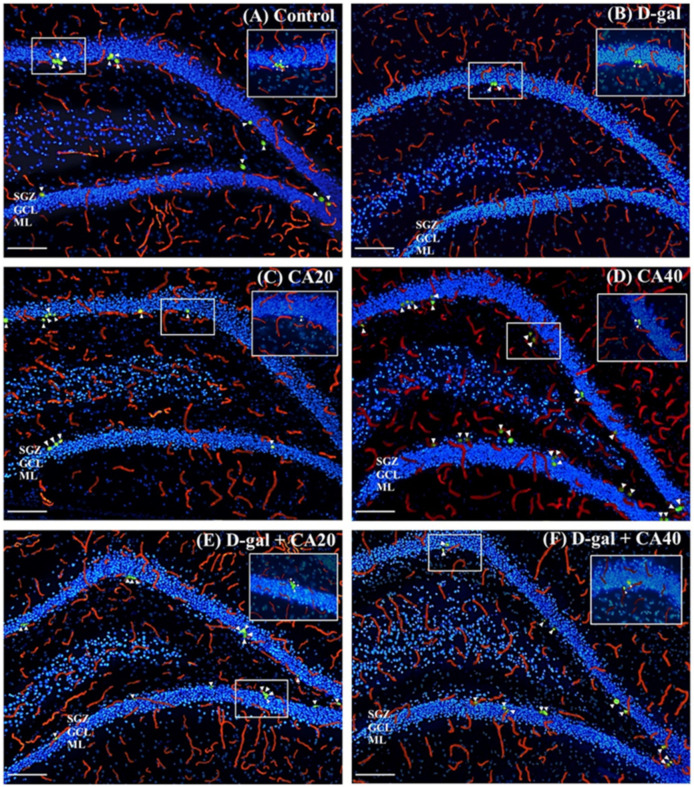
RECA-1/Ki-67 immunofluorescence double staining (**A**–**F**). The control, CA20, CA40, D-gal + CA20 and D-gal + CA40 groups revealed a significantly greater number of blood vessel associated Ki-67 (**G**), non-blood vessel associated Ki-67 (**H**) and total Ki-67 positive cells (**I**) than those in the D-gal group (* *p* < 0.05, ** *p* < 0.01, *** *p* < 0.001 analyzed with the D-gal group). Moreover, administration of CA40 mg/kg was associated with a significantly greater number of blood vessel associated Ki-67 positive cells than in the control group (^#^
*p* < 0.05, ^##^
*p* < 0.05 analyzed with the control group, (**G**)).

**Figure 6 nutrients-14-02169-f006:**
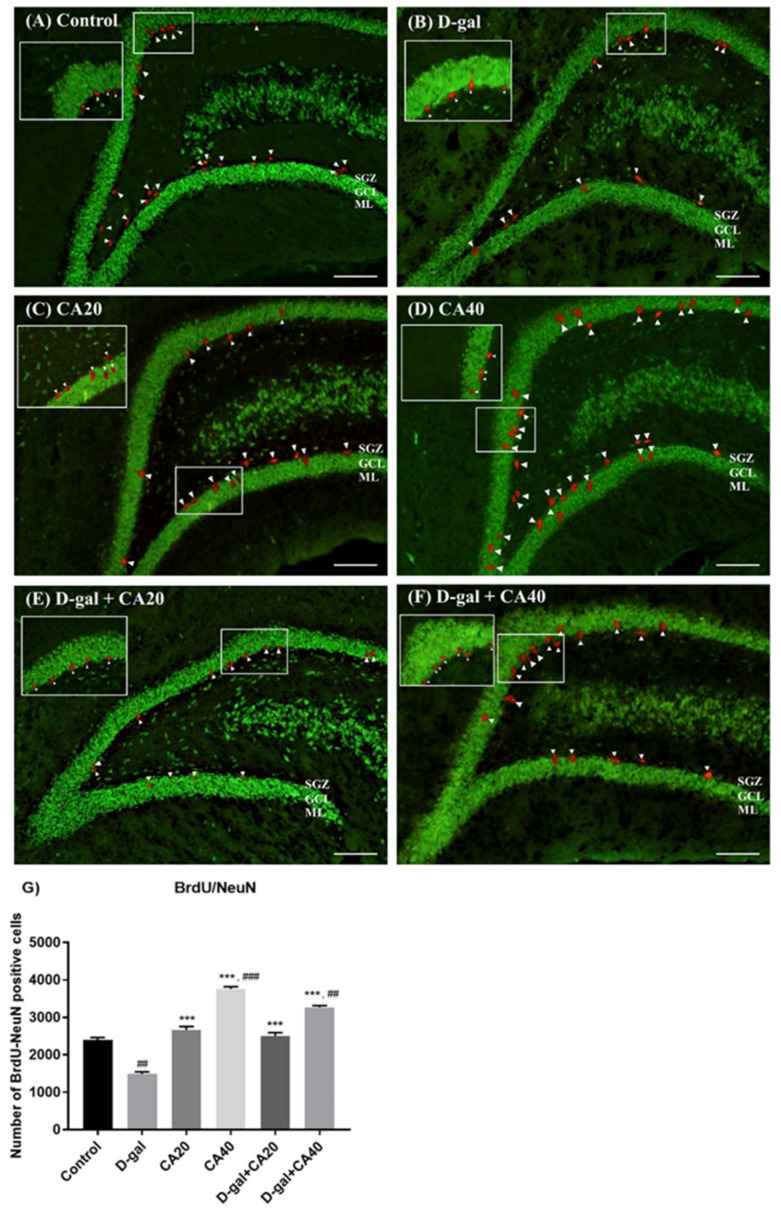
BrdU/NeuN immunofluorescence double staining in all groups (**A**–**F**). The control, CA20, CA40, D-gal + CA20, and D-gal + CA40 groups showed significantly more BrdU/NeuN positive cells than the D-gal group (*** *p* < 0.0001 considered with the D-gal group, (**G**)). D-gal treated rats had a significantly lower BrdU/NeuN positive cell number than the control group (^##^
*p* < 0.01 analyzed with the control group, (**G**)). Moreover, administration of CA40 mg/kg revealed significantly more BrdU/NeuN positive cells than in the control group (^##^
*p* < 0.01, ^###^
*p* < 0.001 considered with the control group, (**G**)).

**Figure 7 nutrients-14-02169-f007:**
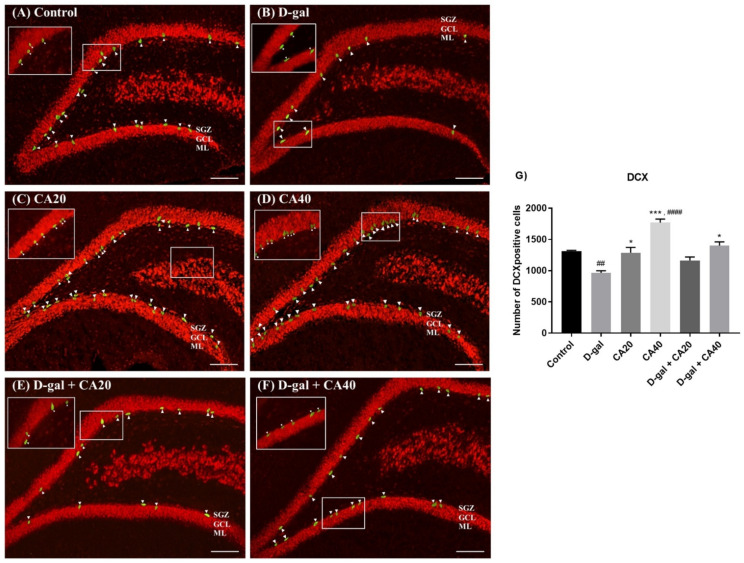
Examination of DCX expression in all groups (**A**–**F**). The control, CA20, CA40 and D-gal + CA40 groups exhibited significantly more DCX positive cells than the D-gal group (* *p* < 0.05, *** *p* < 0.001 considered with the D-gal group, (**G**)). D-gal treatment induced a significant decline of DCX positive cells in comparison with the control group (^##^
*p* < 0.01 considered with the control group, (**G**)). However, DCX positive cells detected after co-administration with CA40 mg/kg were significantly superior to the control group (^####^
*p* < 0.0001 considered with the control group, (**G**)).

**Table 1 nutrients-14-02169-t001:** Distance moved (mean ± SEM) in the novel object location (NOL) test after treatment.

Groups	Distance Moved (cm)
Control	6655 ± 629.3
D-gal	6789 ± 419.0
CA20	7271 ± 519.5
CA40	6792 ± 483.8
D-gal + CA20	5733 ± 584.2
D-gal + CA40	6640 ± 505.0

**Table 2 nutrients-14-02169-t002:** Distance moved (mean ± SEM) in the novel object recognition (NOR) test after treatment.

Groups	Distance Moved (cm)
Control	7110 ± 303.4
D-gal	6890 ± 865.6
CA20	7482 ± 554.4
CA40	6302 ± 569.2
D-gal + CA20	5585 ± 360.0
D-gal + CA40	6615 ± 660.0

## Data Availability

The data examined are openly accessible in source articles and data citations were added in the reference list. The data that help the investigations of this study is also available from the corresponding author, J.U.W.

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
