# Peer review of "Caffeic Acid Alleviates Memory and Hippocampal Neurogenesis Deficits in Aging Rats Induced by D-Galactose"

_nutrients, 2022, doi:10.3390/nu14102169_

Round 1

Reviewer 1 Report

The manuscript is very interesting. I recommend a minor revision only. I suggest to discuss in the "discussion" part the following problems. Are there any proofs that caffeic acid penetrates through blood-brain barrier (BBB)? D-galactose increases the permeability of blood-brain barrier, which can cause that the caffeic acid should enter the brain easier. If the BBB was tight would there be the same effect as the results presented in the manuscript?

 1. Lei M, Zhu Z, Wen Z, Ke S. Impairments of tight junctions are involved in

D-galactose-induced brain aging. Neuroreport. 2013 Aug 21;24(12):671-6. doi:

10.1097/WNR.0b013e3283638f75. PMID: 23820738.

Author Response

Response to Reviewers

Reviewer #1

  1. Are there any proofs that caffeic acid penetrates through blood-brain barrier (BBB)? D-galactose increases the permeability of blood-brain barrier, which can cause that the caffeic acid should enter the brain easier. If the BBB was tight would there be the same effect as the results presented in the manuscript?

After absorption, caffeic acid is transported to three main processes of enzymatic conjugation (known as detoxification) including methylation, sulphation, and glucuronidation via the action of sulfotransferase enzymes, UDP-glucotransferases and catechol-o-methyltransferases, respectively. This makes the compound more hydrophilic that reduces its toxicity and facilitates its elimination (Espíndola et al., 2019). Caffeic acid could not be transported through the blood-brain barrier (BBB) by simple or facilitated diffusion. The transmembrane flow of caffeic acid into intestinal cells occurs through active transport mediated by monocarboxylic acid transporters (MCT) (Espíndola et al., 2019). Numerous solute carriers such as monocarboxylic acid transporters are present on the capillary endothelial cells of BBB (NaÅ‚Ä™cz, 2017). Previous studies suggested that transportation of phenolics is affected by their methylation, sulphatation or glucuronidation (Suominen et al., 2015; Faria et al., 2012). Methylation or glucuronidation of selected polyphenols increases their traversing through BBB models in in vitro study (Faria et al., 2012). Sulphatation of dopamine causes its permeation through the BBB in rats (Suominen et al., 2015).

A previous study confirmed that chronic D-gal administration can cause oxidative stress by increased MDA levels, and decreased T-SOD, GSH-Px, and T-AOC in the hippocampus. Enhancing of oxidative stress may be a main factor for the D-gal-induced brain aging (Lei et al., 2013). Tight junctions are the important structural and functional bases that maintain the integrity of BBB. Lei and co-workers investigated the tight junction in the hippocampus by evaluating levels of ZO-1, occludin, and claudin-5 expression (Lei et al., 2013). Occludin is involved in tight junction strands formation. Loss of occluding expression resulting to the BBB dysfunction (Sheth et al., 2003). ZO-1 is a member of the membrane-associated guanylate kinase family, which plays an important role in the connecting transmembrane proteins and cytoskeleton proteins. Decreasing of ZO-1 may lead to paracellular permeability impairment (Zhao et al., 2011). Claudin-5 is the major component of tight junctions, down-regulation of claudin-5 leading to increased vascular permeability (Luissint et al., 2012). Previous study confirmed that D-gal administration reduced immunoreactivities of ZO-1, occludin, and claudin-5, which disrupt tight junctions (Lei et al., 2013). Down regulation of tight junction protein is related to BBB permeability in brain aging (Simpson et al., 2010). Changes in tight junction associated proteins in structure and function directly affect the integrity of tight junction, leading to the increased permeability of the BBB (Jiao et al., 2011).

The increased permeability of the BBB by D-gal administration may cause caffeic acid to enter the brain easier, which is the same effect as the results presented in our manuscript.

Espíndola KMM, Ferreira RG, Narvaez LEM, Rosario ACRS, da Silva AHM, Silva AGB, et al. Chemical and pharmacological aspects of caffeic acid and its activity in hepatocarcinoma. Front Oncol 2019; 9: 541.

Nałęcz KA. Solute carriers in the blood-brain Barier: safety in abundance. Neurochem Res 2017; 42(3): 795-809.

Suominen T, Piepponen TP, Kostiainen R. Permeation of dopamine sulfate through the blood-brain barrier. PLoS ONE. 2015; 10: e0133904.

Faria A, Mateus N, Calhau C. Flavonoid transport across blood-brain barrier: Implication for their direct neuroprotective actions. Nutr Aging 2012; 1: 89-97.

Lei M, Zhu Z, Wen Z, Ke S. Impairments of tight junctions are involved in D-galactose-induced brain aging. Neuroreport 2013; 24(12): 671-6.

Sheth P, Basuroy S, Li C, Naren AP, Rao RK. Role of phosphatidylinositol 3-kinase in oxidative stress-induced disruption of tight junctions. J Biol Chem 2003; 278: 49239-45.

Zhao J, Wang J, Dong L, Shi H, Wang Z, Ding H, et al. A protease inhibitor against acute stress-induced visceral hypersensitivity and paracellular permeability in rats. Eur J Pharmacol 2011; 654: 289-94.

Luissint AC, Artus C, Glacial F, Ganeshamoorthy K, Couraud PO. Tight junctions at the blood brain barrier: physiological architecture and disease-associated dysregulation. Fluids Barriers CNS 2012; 9: 23.

Simpson JE, Wharton SB, Cooper J, Gelsthorpe C, Baxter L, Forster G, et al. Alterations of the blood–brain barrier in cerebral white matter lesions in the ageing brain. Neurosci Lett 2010; 486: 246-51.

Jiao H, Wang Z, Liu Y, Wang P, Xue Y. Specific role of tight junction proteins claudin-5, occludin, and ZO-1 of the blood–brain barrier in a focal cerebral ischemic insult. J Mol Neurosci 2011; 44: 130-9.

Reviewer #3

  1. The main issue with the manuscript is that there are many ortographic and syntactic errors that should be corrected. English must be thoroughly revised. For instance, titles of paragraphs in the results section should be "D-Gal induced memory impairment in NOL and NOR test is reversed by CA" or "CA reverses D-gal dependent memory impairment" instead of "the results of..".

We have changed the title of paragraphs in the results section as follow:

3.1 D-gal induced memory impairment in NOL test is reversed by CA

3.2 D-gal induced memory impairment in NOR test is reversed by CA

3.3 D-gal reduced Cell Proliferation in the Hippocampal Neurogenesis is reversed by CA

3.4 D-gal reduced Cell Survival related to Mature Neurons in Hippocampus is reversed by CA

3.5 D-gal reduced Immature Neurons in the Hippocampus is reversed by CA

  1. Another point is that the authors stress a lot the idea, also by citing different works, that D-Gal induces oxidative stress and that oxidative stress may be responsible for the observed damage. However, no measures of oxidative stress are shown, not in the D-Gal or in the D-Gal+CA conditions. In my opinion, if the authors could provide a measure of oxidative stress in slices or brain tissue of rats treated with D-Gal and D-Gal+CA the manuscript may be further improved in quality. Furthermore, the red flourescence signal of Fig.7 is not specified.

In this study, we aimed to assess the protective effect of CA on impairment of memory and hippocampal neurogenesis in aged rats induced by D-gal. Our previous study revealed that D-gal administration for 8 weeks resulting in memory impairment and reduction in hippocampal neurogenesis by decreasing levels of Ki-67, BrdU and DCX expression in the subgranular zone of the dentate gyrus in hippocampus (Prajit et al., 2020). Nevertheless, CA shows neuroprotective property in the brain by ameliorating learning and memory ability in high-fat feeding-induced hyperinsulinemia rats (Chang et al., 2019). Therefore, this study aimed to investigate protective effect of CA on impairment of memory and hippocampal neurogenesis in aged rats induced by D-gal.

D-gal can induce brain aging in several mechanisms such as increases of oxidative stress, induction of mitochondrial dysfunction, apoptosis and brain inflammation. When D-gal concentration is increased, it is oxidized by galactose oxidase to form H2O2 resulting in SOD downregulation. The H2O2 forms hydroxide ions (OH-) by reacting to a reduced form of iron. Both H2O2 and OH- are types of ROS, which result in lipid peroxidation in cell membranes and impair redox homeostasis, leading to neuronal damage (Hsieh et al., 2009). Moreover, D-gal acts with amines to form Schiff’s base product (an unstable compound), which followed by the Amadori product (more stable compound) formation and then it is converted to AGE over months or years (Ansari & Dash, 2013; Golubev et al., 2017; Hsieh et al., 2009). After binding of AGE with receptors RAGE, nicotinamide adenine dinucleotide phosphate (NADPH) oxidase and ROS production are increased, leading to neuronal damage and cognitive dysfunction (Hsieh et al., 2009).

For further study, therefore, we plan to investigate the effects of CA on the level of oxidative stress marker such as malondialdehyde (MDA) and antioxidant enzyme activities including superoxide dismutase (SOD), catalase (CAT) and glutathione peroxidase (GPx) in D-gal-induced aging rats.

Prajit R, Sritawan N, Suwannakot K, Naewla S, Aranarochana A, Sirichoat A, et al. Chrysin protects against memory and hippocampal neurogenesis depletion in D-galactose-induced aging in rats. Nutrients 2020; 12: 1100.

Chang W, Huang D, Lo YM, Tee Q, Kuo P, Swibea J, et al. Protective effect of caffeic acid against Alzheimer’s disease pathogenesis via modulating cerebral insulin signaling, β-amyloid accumulation, and synaptic plasticity in hyperinsulinemic rats. J Agric Food Chem 2019; 67: 7684-93.

Hsieh HM, Wu WM, Hu ML. Soy isoflavones attenuate oxidative stress and improve parameters related to aging and Alzheimer's disease in C57BL/6J mice treated with D-galactose. Food Chem Toxicol 2009; 47(3): 625-32.

Ansari NA, Dash D. Amadori glycated proteins: role in production of autoantibodies in diabetes mellitus and effect of inhibitors on non-enzymatic glycation. Aging Dis 2013; 4(1): 50-6.

Golubev A, Hanson AD, Gladyshev VN. Non-enzymatic molecular damage as a prototypic driver of aging. J Biol Chem 2017; 292(15): 6029-38.

We have improved the red fluorescence signal of Fig.7 as follow:

Figure 7. Examination of DCX expression in all groups (A-F)

  1. Finally, I wouldn't stress too much the concept of "aging rats", as two months old animals treated for two months reach 4 months of age, during which they could be simply considered as young adults. Aging, in my view, regards more advanced ages, such as over 9-12 months old, where the animal could be compared to a middle age person.

The laboratory animals such as rats are used in several models of biomedical research including neurobehavioral, cancer and toxicology studies (Sengupta, 2012). Numerous methods have been used to correlate the ages of small mammals with humans such as using the weight of the eye lens (Hardy et al., 1983), growth of molar teeth (Pankakoski, 1980), counting of endosteal layers in the tibia (Broughton et al., 2002), and musculoskeletal growth along with the closure and thickening of the epiphyses (Kahana et al., 2003). These techniques are relative methods and do not exactly define the absolute age.

The most elementary protocol of brain aging is using the aged animals. Eighteen months old male Sprague Dawley rats present neurogenesis and memory impairments (Wu et al., 2012). According to these studies, the period of animal lifespan used in brain aging studies is between 18 and 20 months old, which has no negative effect on locomotor activity. Moreover, D-gal is also used to induce brain aging in animal models (Prajit et al., 2020; Nam et al., 2019; Banji et al., 2013). Male Sprague-Dawley rats (8 weeks old) received D-gal (50 mg/kg) by intraperitoneal injection for 8 weeks showed memory impairment and decreases in cell proliferation, cell survival, and immature neurons (Prajit et al., 2020). Male C57BL/6 mice (5 weeks old) were subcutaneously administered with D-gal (150 mg/kg/day) for 10 weeks, demonstrating that long term D-gal treatment decreases expressions of synaptic plasticity-related indicators, i.e., sanaptophysin, phosphorylated Ca2+/calmodulin-dependent protein kinase II and BDNF in hippocampus. Moreover, long term treatment of D-gal also shows ROS production and neuroinflammation in hippocampus and then causes memory loss (Nam et al., 2019). Nevertheless, doses of D-gal induced brain aging do not affect the locomotor activity (Prajit et al., 2020) and body or brain weight gain of animals (Nam et al., 2019). Moreover, D-gal was used to induce aging on cardiovascular or reproductive systems in different doses and duration of treatment. D-gal administration (50-150 mg/kg/day) by intraperitoneal injection showed negative effect on cardiac senescence markers by increasing AGE protein level, senescence-associated β-galactosidase (SA- β-gal) expression, p21 and p53 protein expression. These suggested that D-gal administration increase cardiac senescence in D-gal-induced aged rats by showing similarities in senescence protein levels with naturally aged rats (Cebe et al., 2014; Wu et al., 2017; Chang et al., 2017). D-gal administration with dose 100-200 mg/kg/day by intraperitoneal injection for 6-8 weeks increased SOD serum level, decreased testicular weight and sperm parameter (Liao et al., 2016). The results confirmed that D-gal administration induced reproductive system aging by demonstrating similarities in characteristics of reproductive organs with naturally aged rat.

Sengupta P. Environmental and occupational exposure of metals and their role in male reproductive functions. Drug Chem Toxicol 2012; 36: 353-68.

Pankakoski E. An improved method for age determination in the muskrat, Ondatra zibethica (L.). Ann Zool Fennici 1980; 17: 113-21.

Broughton JM, Rampton D, Holanda K. A test of an osteologically based age determination technique in the Double crested Cormorant Phalacrocorax autitus. Ibis 2002; 144: 143-6.

Kahana T, Birkby WH, Goldin L, Hiss J. Estimation of age in adolescents: The basilar synchondrosis. J Forensic Sci 2003; 48: 1-5.

Wu B, Chen Y, Huang J, Ning Y, Bian Q, Shan Y, et al., Icariin improves cognitive deficits and activates quiescent neural stem cells in aging rats. J Ethnopharmacol 2012; 142: 746-53.

Prajit R, Sritawan N, Suwannakot K, Naewla S, Aranarochana A, Sirichoat A, et al. Chrysin protects against memory and hippocampal neurogenesis depletion in D-galactose-induced aging in rats. Nutrients 2020; 12: 1100.

Nam SM, Seo M, Seo JS, Rhim H, Nahm SS, Cho IH, et al. Ascorbic acid mitigates D-galactose-induced brain aging by increasing hippocampal neurogenesis and improving memory function. Nutrients 2019; 11: 176.

Banji D, Banji OJ, Dasaroju S, Kranthi KC. Curcumin and piperine abrogate lipid and protein oxidation induced by D-galactose in rat brain. Brain Res 2013; 1515: 1-11.

Cebe T, Yanar K, Atukeren P, Ozan T, Kuruç AI, Kunbaz A et al. A comprehensive study of myocardial redox homeostasis in naturally and mimetically aged rats. Age 2014; 36: 9728.

Wu W, Hou CL, Mu XP, Sun C, Zhu YC, Wang MJ et al. H2S Donor NaHS changes the production of endogenous H2S and NO in D-galactose-induced accelerated ageing. Oxid Med Cell Longev. 2017; 2017: 5707830. 8.

Chang YM, Chang HH, Lin HJ, Tsai CC, Tsai CT, Chang HN et al. Inhibi-tion of cardiac hypertrophy effects in D-galactose-induced senescent hearts by alpinate oxyphyllae fructus treatment. Evid Based Complement Alternat Med. 2017; 2017: 2624384.

Liao CH, Chen BH, Chiang HS, Chen CW, Chen MF, Ke CC et al. Optimizing a Male Reproductive Aging MouseModel by D-Galactose Injection. Int J Mo Sci 2016; 17: 1-10.

Reviewer 2 Report

In this manuscript, the authors aim to establish a relationship between caffeic acid consumption, and the memory and neurogenesis processes associated with the aging process.

The manuscript is generally well written and clear, with well-detailed methods, it does however have some minor issues that I would like to see clarified before publication. I also suggest the correction of some minor English inaccuracies (e.g lines 42 and 155).

The legends in the figures of immunohistochemistry should be bigger to be possible to read.

Please, define what you meant with positive cells in line 143 of the manuscript.

Please explain better the physiological relevance of the ingestion of D-Gal or if it is only used to establish the mouse model in use.

Author Response

Reviewer #2

  1. The manuscript is generally well written and clear, with well-detailed methods, it does however have some minor issues that I would like to see clarified before publication. I also suggest the correction of some minor English inaccuracies (e.g lines 42 and 155).

We have corrected English wording in lines 42 and 155 that show text changed in manuscript

  1. The legends in the figures of immunohistochemistry should be bigger to be possible to read.

We have improved the legends in the figures of immunohistochemistry (figure 5,6 and 7) as follows.

Figure 5. RECA-1/Ki-67 immunofluorescence double staining (A-F)

Figure 6. BrdU/NeuN immunofluorescence double staining in all groups (A-F).

Figure 7. Examination of DCX expression in all groups (A-F)

  1. Please, define what you meant with positive cells in line 143 of the manuscript.

            - We have defined the positive cells in line 143 as the DCX and BrdU/NeuN positive cells as shown in the manuscript with text changed

  1. Please explain better the physiological relevance of the ingestion of D-Gal or if it is only used to establish the mouse model in use.

In this study, D-gal is purposely used to induce aging in a rat model. For the physiological mechanism, Carbohydrates are digested by amylases from salivary and pancreas. Then, it is converted to monosaccharides (glucose, fructose and galactose) by enzymes in the brush border membrane of enterocytes (the epithelial cells in the intestine). After that, galactose and fructose are transferred through the intestine wall to the hepatic portal vein and then to parenchymal cells of the liver. Fructose and galactose are phosphorylated by fructokinase and galactokinase, respectively. These specific enzymes are found only in the liver and changed to glucose. Finally, an important energy from glucose is moved from the liver through the bloodstream to supply all the body cells (Bjelakovic et al., 2011).

            The main pathway of galactose metabolism is Lelior pathway, which initiates in the liver. A disaccharide of galactose and glucose is lactose, which is digested by lactase into its elemental monosaccharides in the brush border membranes of the enterocytes. The sodium/glucose cotransporter 1 (SGLT1) is required to transport galactose across the brush border membrane. Firstly, a sodium ion binds to the outer surface of the transporter leading to galactose binding. Then, galactose is released into the cytoplasm in the inner surface and followed by releasing of sodium ions (Wright, 2013). Then, galactose is transported across the basolateral membrane by glucose transporter 2 (GLUT2). After efflux from the enterocyte by exocytosis, galactose enters the portal vein and is transported into the liver (Augustin, 2010; Leturque et al., 2009; Bjelakovic et al., 2011).

In normal condition, D-gal is metabolized into glucose by galactokinase and uridyl transferase, which enters the glycolysis pathway or is stored as glycogen in liver, muscle and adipose tissue (Coelho et al., 2015). Normally, concentration of D-gal in blood is less than 10 mg/dL (Shwe et al., 2018). The maximal concentration of galactose recommended daily for healthy adults is 50 g, beyond this dose can be eliminated from the body within 8 hours after ingestion (Morava, 2014). Excessive amounts of D-gal can generate ROS, leading to mitochondrial dysfunction, oxidative stress, inflammation and apoptosis, especially in neuronal cells (Shwe et al., 2018).

In mouse model, D-gal can induce brain aging in several mechanisms such as increases of oxidative stress, induction of mitochondrial dysfunction, apoptosis and brain inflammation. When D-gal concentration is enhanced, it is oxidized by galactose oxidase to form H2O2 resulting in SOD downregulation. The H2O2 forms hydroxide ions (OH-) by reacting to a reduced form of iron. Both H2O2 and OH- are types of ROS, which result in lipid peroxidation in cell membranes and impair redox homeostasis, leading to neuronal damage (Hsieh et al., 2009). Moreover, D-gal acts with amines to form Schiff’s base product (an unstable compound), which followed by the Amadori product (more stable compound) formation and then, it is converted to AGE over months or years (Ansari & Dash, 2013; Golubev et al., 2017; Hsieh et al., 2009). After binding of AGE with receptors RAGE, nicotinamide adenine dinucleotide phosphate (NADPH) oxidase and ROS production are increased, leading to neuronal damage and cognitive dysfunction (Hsieh et al., 2009). Several studies reported that D-gal induces brain aging by increasing oxidative stress in various brain regions including the hippocampus, cerebral cortex, and auditory cortex. D-gal has been used by varying doses between 50 mg/kg/day to 500 mg/kg/day and periods of administration time between 6-8 weeks (Prajit et al., 2020; Banji et al., 2013; Chen et al., 2011; Du et al., 2012; Long et al., 2007).

Increasing effects of D-gal on apoptosis occurs in both extrinsic and intrinsic pathways. For the extrinsic pathway, it activates effector caspases directly through JNK (c-Jun-N-terminal kinase) and gathers with the intrinsic apoptotic pathway at the mitochondrion (Benn & Woolf, 2004). Furthermore, D-gal motivates the mitochondria to release cytochrome c, reduces the levels of anti-apoptotic Bcl2 expression and increases apoptotic Bax (Qian et al., 2008). A previous study has reported that long-term treatment of D-gal increases the inflammatory markers such as cyclooxygenase (COX-2), inducible nitric oxide synthase (iNOS), nitric oxide synthase-2 (NOS-2), TNF-α, IL-1β, IL-6, and nuclear factor kappa-light-chain-enhancer of activated B cells (NF-κB) and also induces neuroinflammatory via the activation of the transcription factor NFκ-B through Ras and redox-sensitive signaling pathways, leading to memory deficit. The dosage of D-gal administration to produce inflammation range from 50 to 180 mg/kg/day and treatment period from 6 weeks to 60 days (Shwe et al., 2018; Yang et al., 2016; Yu et al., 2015; Lu et al., 2010; Ullah et al., 2015).

Bjelakovic G, Stojanovic I, Jevtovic-Stoimenov T, Saranac Lj, Bjelakovic B, Pavlovic D, et al. Hypoglycemia as a pathological result in medical praxis. Type 1 Diabetes Complications 2011; Doi: 10.5772/24754.

Wright EM. Glucose transport families SLC5 and SLC50. Mol Aspects Med 2013; 34: 183-96.

Augustin R. The protein family of glucose transport facilitators: it’s not only about glucose after all. IUBMB Life 2010; 62: 315-33.

Leturque A, Brot-Laroche E, Le Gall M. GLUT2 mutations, translocation, and receptor function in diet sugar managing. Am J Physiol Endocrinol Metab 2009; 296: E985-E992.

Coelho AI, Berry GT, Rubio-Gozalbo ME. Galactose metabolism and health. Curr Opin Clin Nutr Metab Care 2015; 18: 422-7.

Shwe T, Pratchayasakul W, Chattipakorn N, Chattipakorn SC. Role of D-galactose-induced brain aging and its potential used for therapeutic interventions. Exp Gerontol 2018; 101: 13-36.

Morava E. Galactose supplementation in phosphoglucomutase-1 deficiency; review and outlook for a novel treatable CDG. Mol Genet Metab 2014; 112: 275-9.

Hsieh HM, Wu WM, Hu ML. Soy isoflavones attenuate oxidative stress and improve parameters related to aging and Alzheimer's disease in C57BL/6J mice treated with D-galactose. Food Chem Toxicol 2009; 47(3): 625-32.

Ansari NA, Dash D. Amadori glycated proteins: role in production of autoantibodies in diabetes mellitus and effect of inhibitors on non-enzymatic glycation. Aging Dis 2013; 4(1): 50-6.

Golubev A, Hanson AD, Gladyshev VN. Non-enzymatic molecular damage as a prototypic driver of aging. J Biol Chem 2017; 292(15): 6029-38.

Prajit R, Sritawan N, Suwannakot K, Naewla S, Aranarochana A, Sirichoat A, et al. Chrysin protects against memory and hippocampal neurogenesis depletion in D-galactose-induced aging in rats. Nutrients 2020; 12: 1100.

Banji D, Banji OJ, Dasaroju S, Kranthi KC. Curcumin and piperine abrogate lipid and protein oxidation induced by D-galactose in rat brain. Brain Res 2013; 1515: 1-11.

Chen B, Zhong Y, Peng W, Sun Y, Hu Yj, Yang Y, et al. Increased mitochondrial DNA damage and decreased base excision repair in the auditory cortex of D-galactose-induced aging rats. Mol Biol Rep 2011; 38: 3635-42.

Du Z, Hu Y, Yang Y, Sun Y,  Zhang S, Zhou T, et al. NADPH oxidase-dependent oxidative stress and mitochondrial damage in hippocampus of D-galactose-induced aging rats. J Huazhong Univ Sci Technolog Med Sci 2012; 32: 466-72.

Long J, Wang X, Gao H, Liu Z, Liu C, Miao M, et al. D-Galactose toxicity in mice is associated with mitochondrial dysfunction: protecting effects of mitochondrial nutrient R-alpha-lipoic acid. Biogerontology 2007; 8: 373-81.

Benn SC, Woolf CJ. Adult neuron survival strategies--slamming on the brakes. Nat Rev Neurosci 2004; 5(9): 686-700.

Qian YF, Wang H, Yao WB, Gao XD. Aqueous extract of the Chinese medicine, Danggui-Shaoyao-San, inhibits apoptosis in hydrogen peroxide-induced PC12 cells by preventing cytochrome c release and inactivating of caspase cascade. Cell Biol Int 2008; 32(2): 304-11.

Yang H, Qu Z, Zhang J, Huo L, Gao J, Gao W. Ferulic acid ameliorates memory impairment in d-galactose-induced aging mouse model. Int J Food Sci Nutr 2016; 67(7): 806-17.

Yu Y, Bai F, Wang W, Liu Y, Yuan Q, Qu S, et al. Fibroblast growth factor 21 protects mouse brain against D-galactose induced aging via suppression of oxidative stress response and advanced glycation end products formation. Pharmacol Biochem Behav 2015; 133: 122-31.

Lu J, Wu DM, Zheng YL, Hu B, Zhang ZF. Purple sweet potato color alleviates D-galactose-induced brain aging in old mice by promoting survival of neurons via PI3K pathway and inhibiting cytochrome C-mediated apoptosis. Brain Pathol. 2010; 20: 598-612.

Ullah F, Ali T, Ullah N, Kim MO. Caffeine prevents d-galactose-induced cognitive deficits, oxidative stress, neuroinflammation and neurodegeneration in the adult rat brain. Neurochem Int 2015; 90: 114-24.

Reviewer 3 Report

The work from Saenno et al. describes the effects of D-Gal supplementation on learning and memory and hippocampal neurogenesis and the effect of CA in halting and reversing the D-Gal-dependent deleterious effects. Experimentally, the work is nicely performed with all the necessary control experiments. Despite the fact that the work is mainly descriptive as it doesn't identify a cause-effect relationship between D-Gal damage and CA-induced rescue I still appreciate its simplicity and linearity.

1) The main issue with the manuscript is that there are many ortographic and syntactic errors that should be corrected. English must be thoroughly revised. For instance, titles of paragraphs in the results section should be "D-Gal induced memory impairment in NOL and NOR test is reversed  by CA" or "CA reverses D-gal dependent memory impairment" instead of "the results of..".

2) Another point is that the authors stress a lot the idea, also by citing different works, that D-Gal induces oxidative stress and that oxidative stress may be responsible for the observed damage. However, no measures of oxidative stress are shown, not in the D-Gal or in the D-Gal+CA conditions. In my opinion, if the authors could provide a measure of oxidative stress in slices or brain tissue of rats treated with D-Gal and D-Gal+CA the manuscript may be further improved in quality. Furthermore, the red flourescence signal of Fig.7 is not specified.

3) Finally, I wouldn't stress too much the concept of "aging rats", as two months old animals treated for two months reach 4 months of age, during which they could be simply considered as young adults. Aging, in my view, regards more advanced ages, such as over 9-12 months old, where the animal could be compared to a middle age person.

Author Response

Reviewer #3

  1. The main issue with the manuscript is that there are many ortographic and syntactic errors that should be corrected. English must be thoroughly revised. For instance, titles of paragraphs in the results section should be "D-Gal induced memory impairment in NOL and NOR test is reversed by CA" or "CA reverses D-gal dependent memory impairment" instead of "the results of..".

We have changed the title of paragraphs in the results section as follow:

3.1 D-gal induced memory impairment in NOL test is reversed by CA

3.2 D-gal induced memory impairment in NOR test is reversed by CA

3.3 D-gal reduced Cell Proliferation in the Hippocampal Neurogenesis is reversed by CA

3.4 D-gal reduced Cell Survival related to Mature Neurons in Hippocampus is reversed by CA

3.5 D-gal reduced Immature Neurons in the Hippocampus is reversed by CA

  1. Another point is that the authors stress a lot the idea, also by citing different works, that D-Gal induces oxidative stress and that oxidative stress may be responsible for the observed damage. However, no measures of oxidative stress are shown, not in the D-Gal or in the D-Gal+CA conditions. In my opinion, if the authors could provide a measure of oxidative stress in slices or brain tissue of rats treated with D-Gal and D-Gal+CA the manuscript may be further improved in quality. Furthermore, the red flourescence signal of Fig.7 is not specified.

In this study, we aimed to assess the protective effect of CA on impairment of memory and hippocampal neurogenesis in aged rats induced by D-gal. Our previous study revealed that D-gal administration for 8 weeks resulting in memory impairment and reduction in hippocampal neurogenesis by decreasing levels of Ki-67, BrdU and DCX expression in the subgranular zone of the dentate gyrus in hippocampus (Prajit et al., 2020). Nevertheless, CA shows neuroprotective property in the brain by ameliorating learning and memory ability in high-fat feeding-induced hyperinsulinemia rats (Chang et al., 2019). Therefore, this study aimed to investigate protective effect of CA on impairment of memory and hippocampal neurogenesis in aged rats induced by D-gal.

D-gal can induce brain aging in several mechanisms such as increases of oxidative stress, induction of mitochondrial dysfunction, apoptosis and brain inflammation. When D-gal concentration is increased, it is oxidized by galactose oxidase to form H2O2 resulting in SOD downregulation. The H2O2 forms hydroxide ions (OH-) by reacting to a reduced form of iron. Both H2O2 and OH- are types of ROS, which result in lipid peroxidation in cell membranes and impair redox homeostasis, leading to neuronal damage (Hsieh et al., 2009). Moreover, D-gal acts with amines to form Schiff’s base product (an unstable compound), which followed by the Amadori product (more stable compound) formation and then it is converted to AGE over months or years (Ansari & Dash, 2013; Golubev et al., 2017; Hsieh et al., 2009). After binding of AGE with receptors RAGE, nicotinamide adenine dinucleotide phosphate (NADPH) oxidase and ROS production are increased, leading to neuronal damage and cognitive dysfunction (Hsieh et al., 2009).

For further study, therefore, we plan to investigate the effects of CA on the level of oxidative stress marker such as malondialdehyde (MDA) and antioxidant enzyme activities including superoxide dismutase (SOD), catalase (CAT) and glutathione peroxidase (GPx) in D-gal-induced aging rats.

Prajit R, Sritawan N, Suwannakot K, Naewla S, Aranarochana A, Sirichoat A, et al. Chrysin protects against memory and hippocampal neurogenesis depletion in D-galactose-induced aging in rats. Nutrients 2020; 12: 1100.

Chang W, Huang D, Lo YM, Tee Q, Kuo P, Swibea J, et al. Protective effect of caffeic acid against Alzheimer’s disease pathogenesis via modulating cerebral insulin signaling, β-amyloid accumulation, and synaptic plasticity in hyperinsulinemic rats. J Agric Food Chem 2019; 67: 7684-93.

Hsieh HM, Wu WM, Hu ML. Soy isoflavones attenuate oxidative stress and improve parameters related to aging and Alzheimer's disease in C57BL/6J mice treated with D-galactose. Food Chem Toxicol 2009; 47(3): 625-32.

Ansari NA, Dash D. Amadori glycated proteins: role in production of autoantibodies in diabetes mellitus and effect of inhibitors on non-enzymatic glycation. Aging Dis 2013; 4(1): 50-6.

Golubev A, Hanson AD, Gladyshev VN. Non-enzymatic molecular damage as a prototypic driver of aging. J Biol Chem 2017; 292(15): 6029-38.

We have improved the red fluorescence signal of Fig.7 as follow:

Figure 7. Examination of DCX expression in all groups (A-F)

  1. Finally, I wouldn't stress too much the concept of "aging rats", as two months old animals treated for two months reach 4 months of age, during which they could be simply considered as young adults. Aging, in my view, regards more advanced ages, such as over 9-12 months old, where the animal could be compared to a middle age person.

The laboratory animals such as rats are used in several models of biomedical research including neurobehavioral, cancer and toxicology studies (Sengupta, 2012). Numerous methods have been used to correlate the ages of small mammals with humans such as using the weight of the eye lens (Hardy et al., 1983), growth of molar teeth (Pankakoski, 1980), counting of endosteal layers in the tibia (Broughton et al., 2002), and musculoskeletal growth along with the closure and thickening of the epiphyses (Kahana et al., 2003). These techniques are relative methods and do not exactly define the absolute age.

The most elementary protocol of brain aging is using the aged animals. Eighteen months old male Sprague Dawley rats present neurogenesis and memory impairments (Wu et al., 2012). According to these studies, the period of animal lifespan used in brain aging studies is between 18 and 20 months old, which has no negative effect on locomotor activity. Moreover, D-gal is also used to induce brain aging in animal models (Prajit et al., 2020; Nam et al., 2019; Banji et al., 2013). Male Sprague-Dawley rats (8 weeks old) received D-gal (50 mg/kg) by intraperitoneal injection for 8 weeks showed memory impairment and decreases in cell proliferation, cell survival, and immature neurons (Prajit et al., 2020). Male C57BL/6 mice (5 weeks old) were subcutaneously administered with D-gal (150 mg/kg/day) for 10 weeks, demonstrating that long term D-gal treatment decreases expressions of synaptic plasticity-related indicators, i.e., sanaptophysin, phosphorylated Ca2+/calmodulin-dependent protein kinase II and BDNF in hippocampus. Moreover, long term treatment of D-gal also shows ROS production and neuroinflammation in hippocampus and then causes memory loss (Nam et al., 2019). Nevertheless, doses of D-gal induced brain aging do not affect the locomotor activity (Prajit et al., 2020) and body or brain weight gain of animals (Nam et al., 2019). Moreover, D-gal was used to induce aging on cardiovascular or reproductive systems in different doses and duration of treatment. D-gal administration (50-150 mg/kg/day) by intraperitoneal injection showed negative effect on cardiac senescence markers by increasing AGE protein level, senescence-associated β-galactosidase (SA- β-gal) expression, p21 and p53 protein expression. These suggested that D-gal administration increase cardiac senescence in D-gal-induced aged rats by showing similarities in senescence protein levels with naturally aged rats (Cebe et al., 2014; Wu et al., 2017; Chang et al., 2017). D-gal administration with dose 100-200 mg/kg/day by intraperitoneal injection for 6-8 weeks increased SOD serum level, decreased testicular weight and sperm parameter (Liao et al., 2016). The results confirmed that D-gal administration induced reproductive system aging by demonstrating similarities in characteristics of reproductive organs with naturally aged rat.

Sengupta P. Environmental and occupational exposure of metals and their role in male reproductive functions. Drug Chem Toxicol 2012; 36: 353-68.

Pankakoski E. An improved method for age determination in the muskrat, Ondatra zibethica (L.). Ann Zool Fennici 1980; 17: 113-21.

Broughton JM, Rampton D, Holanda K. A test of an osteologically based age determination technique in the Double crested Cormorant Phalacrocorax autitus. Ibis 2002; 144: 143-6.

Kahana T, Birkby WH, Goldin L, Hiss J. Estimation of age in adolescents: The basilar synchondrosis. J Forensic Sci 2003; 48: 1-5.

Wu B, Chen Y, Huang J, Ning Y, Bian Q, Shan Y, et al., Icariin improves cognitive deficits and activates quiescent neural stem cells in aging rats. J Ethnopharmacol 2012; 142: 746-53.

Prajit R, Sritawan N, Suwannakot K, Naewla S, Aranarochana A, Sirichoat A, et al. Chrysin protects against memory and hippocampal neurogenesis depletion in D-galactose-induced aging in rats. Nutrients 2020; 12: 1100.

Nam SM, Seo M, Seo JS, Rhim H, Nahm SS, Cho IH, et al. Ascorbic acid mitigates D-galactose-induced brain aging by increasing hippocampal neurogenesis and improving memory function. Nutrients 2019; 11: 176.

Banji D, Banji OJ, Dasaroju S, Kranthi KC. Curcumin and piperine abrogate lipid and protein oxidation induced by D-galactose in rat brain. Brain Res 2013; 1515: 1-11.

Cebe T, Yanar K, Atukeren P, Ozan T, Kuruç AI, Kunbaz A et al. A comprehensive study of myocardial redox homeostasis in naturally and mimetically aged rats. Age 2014; 36: 9728.

Wu W, Hou CL, Mu XP, Sun C, Zhu YC, Wang MJ et al. H2S Donor NaHS changes the production of endogenous H2S and NO in D-galactose-induced accelerated ageing. Oxid Med Cell Longev. 2017; 2017: 5707830. 8.

Chang YM, Chang HH, Lin HJ, Tsai CC, Tsai CT, Chang HN et al. Inhibi-tion of cardiac hypertrophy effects in D-galactose-induced senescent hearts by alpinate oxyphyllae fructus treatment. Evid Based Complement Alternat Med. 2017; 2017: 2624384.

Liao CH, Chen BH, Chiang HS, Chen CW, Chen MF, Ke CC et al. Optimizing a Male Reproductive Aging MouseModel by D-Galactose Injection. Int J Mo Sci 2016; 17: 1-10.
